# Early Life Short-Term Exposure to Polychlorinated Biphenyl 126 in Mice Leads to Metabolic Dysfunction and Microbiota Changes in Adulthood

**DOI:** 10.3390/ijms23158220

**Published:** 2022-07-26

**Authors:** Yuan Tian, Bipin Rimal, Wei Gui, Imhoi Koo, Shigetoshi Yokoyama, Gary H. Perdew, Andrew D. Patterson

**Affiliations:** 1Department of Veterinary and Biomedical Sciences, The Pennsylvania State University, University Park, PA 16802, USA; yzt11@psu.edu (Y.T.); bur157@psu.edu (B.R.); iuk41@psu.edu (I.K.); sqy5346@psu.edu (S.Y.); ghp2@psu.edu (G.H.P.); 2Huck Institutes of the Life Sciences, The Pennsylvania State University, University Park, PA 16802, USA; wug28@psu.edu

**Keywords:** 3,3′,4,4′,5-pentacholorobiphenyl, early life exposure, metabolomics, gut microbiota, metabolic disorders

## Abstract

Early life exposure to environmental pollutants may have long-term consequences and harmful impacts on health later in life. Here, we investigated the short- and long-term impact of early life 3,3′,4,4′,5-pentacholorobiphenyl (PCB 126) exposure (24 μg/kg body weight for five days) in mice on the host and gut microbiota using 16S rRNA gene sequencing, metagenomics, and ^1^H NMR- and mass spectrometry-based metabolomics. Induction of *Cyp1a1*, an aryl hydrocarbon receptor (AHR)-responsive gene, was observed at 6 days and 13 weeks after PCB 126 exposure consistent with the long half-life of PCB 126. Early life, Short-Term PCB 126 exposure resulted in metabolic abnormalities in adulthood including changes in liver amino acid and nucleotide metabolism as well as bile acid metabolism and increased hepatic lipogenesis. Interestingly, early life PCB 126 exposure had a greater impact on bacteria in adulthood at the community structure, metabolic, and functional levels. This study provides evidence for an association between early life environmental pollutant exposure and increased risk of metabolic disorders later in life and suggests the microbiome is a key target of environmental chemical exposure.

## 1. Introduction

Throughout life, humans are exposed to a wide range of environmental contaminants in the air, water, and diet. Research has established a relationship between environmental exposure and disease including cancer [1,2], nervous system disorders [2,3], gastrointestinal illness [4,5], cardiovascular diseases [6,7], and metabolic diseases [8,9]. However, it is still unclear how these exposures affect human health, what levels are harmful, and how timing of exposure affects toxicity.

Historically, risk assessment from environmental exposure mainly focused on adults and gave less consideration to vulnerable life stages such as early childhood [10]. Children may have increased exposures to environmental contaminants since they have a higher inhalation rate and a higher body surface area to body weight ratio [11]. Growing evidence supports that early life exposure to environmental pollutants has long-term consequences and has been linked to disease later in life [12,13]. Therefore, data on developmental toxicity of early life exposures are needed.

3,3′,4,4′,5-pentacholorobiphenyl (PCB 126), one of the most acutely toxic planar dioxin-like polychlorinated biphenyl (PCB) congeners, has a long half-life (4–5 years in humans) [14]. Animal fat tissues (such as meat and dairy products) are a major source of PCBs, since PCBs are lipophilic and highly resistant to physical, chemical, and enzymatic breakdown [15]. PCB 126 exposure causes several chronic toxic effects, including cancer development [16,17], hepatic dysfunction [18,19], gastrointestinal illnesses [20,21], and metabolic syndrome [22,23]. Accumulating evidence suggests that early life exposure to PCB 126 can have long-term consequences on behavior and growth [12,13,24]. A recent study reported that exposure of early zebrafish embryos to relatively low doses of PCB 126 (0.3–1.2 nM) altered normal brain development by reprogramming gene expression patterns, which may result in alterations in adult behavior [12,24]. Early life exposure to PCB 126 showed delayed toxicity and resulted in delayed mortality as well as growth impairment and delayed development in the zebrafish model [13]. Recent studies demonstrated that PCB 126 significantly altered the gut microbial ecosystem in adult mice and in in vitro models identifying significant microbial toxicity following environment pollutant exposure [20,25]. However, a systematic investigation of the long-term consequences of early life, Short-Term PCB 126 exposure on host and bacterial metabolism has not been described. Furthermore, there are no regulatory guidelines that address chemical-induced altered microbiome, which may lead to health issues.

Here, we combined 16S rRNA gene sequencing, metagenomics, and ^1^H NMR- and mass spectrometry-based metabolomics to determine the short- and long-term effects of early life Short-Term exposure to PCB 126 on host and gut microbiota in mice. We demonstrate that early life PCB 126 exposure resulted in delayed metabolic abnormalities in adulthood, including significant changes in liver global metabolism and bile acid metabolism as well as increased hepatic lipogenesis. Early life exposure to PCB 126 affected the gut microbiota later in life at the community structure, metabolic, and functional levels. This study provides new metabolic information on the long-term consequences of early life environmental pollutants exposure and finds significant association between early life environmental exposure and abnormal metabolism in adulthood.

## 2. Results

### 2.1. Early Life PCB 126 Exposure Results in Persistently AHR Activation and Oxidative Stress

Short-Term PCB 126 (24 µg/kg body weight for five days) exposure at four weeks old (Appendix A) had no effect on body weight, blood biochemical markers, serum cytokines, or liver histopathology in mice at the 6th day or 13th week after exposure (Appendix A). The mRNA expression of AHR target genes in the liver and ileum were significantly higher at the 6th day and 13th week after PCB 126 exposure (Figure 1A,B), consistent with the long half-life of PCB 126 in the rodents [14]. Significantly higher ratios of oxidized glutathione (GSSG) to reduced glutathione (GSH) in the liver were observed in mice at the 13th week after PCB 126 exposure (Figure 1C).

### 2.2. Early Life PCB 126 Exposure Disrupts Liver Global Metabolism in Adulthood

To explore the influence of early life PCB 126 exposure on host metabolism, liver hydrophilic metabolite profiling was performed using global ^1^H NMR analysis. Five days of PCB 126 exposure rapidly altered hepatic amino acid metabolism, including significantly lower levels of alanine, glutamate, tyrosine, phenylalanine, histidine, and tryptophan in the liver of mice on the 6th day after exposure (Figure 2A). Notably, early life PCB 126 exposure also exhibited a pronounced effect on hepatic amino acid and nucleotide metabolism in mice at the 13th week after exposure (Figure 2B). The significantly lower levels of branched-chain amino acids (BCAAs), alanine, lysine, glutamate, tyrosine, and phenylalanine but higher levels of glutamine, inosine, guanosine, adenosine monophosphate (AMP), adenosine diphosphate (ADP), adenosine triphosphate (ATP), cytidine 5′-monophosphate (CMP), uridine monophosphate (UMP), and uridine 5′-diphosphate (UDP) were observed in the liver of mice with PCB 126 exposure at the 13th week (Figure 2B).

### 2.3. Early Life PCB 126 Exposure Increases Hepatic Lipid Accumulation in Adulthood

Oil red O staining and F4/80 immunohistochemistry of the liver revealed that early life PCB 126 exposure resulted in significant increases in intracellular lipid droplets and infiltrating macrophages (F4/80-positive cells) in mice at the 13th week after exposure (Figure 3A). Hepatic triglyceride quantification also demonstrated that fat accumulation was significantly increased in the liver of mice with PCB 126 exposure at the 13th week (Figure 3B). As further validation, quantitative ^1^H NMR analysis confirmed that PCB 126 exposure resulted in significantly higher levels of hepatic lipids and fatty acids including total cholesterol (TC), free cholesterol (FC), phosphatidylethanolamine (PE), unsaturated fatty acid (UFA), monosaturated fatty acid (MUFA), and polyunsaturated fatty acid (PUFA) in the liver of mice at the 13th week (Figure 3C,D). Consistently, PCB 126 exposure also increased mRNA expression of genes involved in de novo fatty acid biosynthesis in the liver of mice at the 13th week after exposure (Figure 3E). No significant change in lipid profiles or mRNA expression in the liver from mice with PCB 126 exposure on the 6th day was observed (Figure 3B–E).

### 2.4. Early Life PCB 126 Exposure Alters Bile Acid Metabolism in Adulthood

Analysis of bile acid composition showed that early life PCB 126 exposure significantly increased unconjugated and taurine-conjugated bile acids in the liver and fecal samples from mice at the 13th week after exposure (Figure 4B and Appendix A). The mRNA expression of bile acid synthesis, conjugation, and transport enzymes was also significantly higher in the liver of mice with PCB 126 exposure at the 13th week (Figure 4D). We did not observe the substantial changes in bile acid profiling or mRNA expression in mice with PCB 126 exposure on the 6th day (Figure 4A,C and Appendix A).

### 2.5. Early Life PCB 126 Persistently Disrupts Bacteria Composition, Gene Expression, and Metabolism

To investigate the influence of early life PCB 126 exposure on the gut microbiota, 16S rRNA gene sequencing and metagenomics combined with UHPLC-MS/MS and ^1^H NMR-based metabolomics analysis was performed. Early life PCB 126 resulted in significant decrease in relative abundance of *Verrucomicrobiota* phyla in mice at the 13th week after exposure but no significant changes in the major phyla at 6th day after exposure (Appendix A). No significant changes in the ratio of *Firmicutes* to *Bacteroidetes* were observed in mice both at the 6th day and 13th week after PCB 126 exposure (Appendix A). PCB 126 resulted in significant increases in relative abundance of genus *Lachnospiraceae UCG-006* and *Dubosiella* in mice on the 6th day after exposure; and increase in relative abundance of genus *Lachnoclostridium*, but decreases in relative abundance of genus *Akkermansia*, *Dubosiella*, *Tyzzerella*, and *Eubacterium siraeum* group in mice at the 13th week after exposure (Figure 5A).

The KEGG database was used to predict the potential metabolic pathway of gut microbiota. KEGG pathway analysis showed significant changes in relative abundance of bacterial metabolic pathways involved in amino acid metabolism, nucleotide metabolism, and energy metabolism in mice on the 6th day after PCB 126 exposure (Figure 5B). Interestingly, early life PCB 126 resulted in a greater impact on microbial pathway at the 13th week compared to 6th day after exposure (Figure 5B). A total of 58 pathways were significantly changed in mice at the 13th week after PCB 126 exposure, including multiple biosynthesis and metabolic of amino acids and nucleotides as well as lipid metabolism, cell wall biosynthesis, and energy metabolism (Figure 5B).

Having determined the significant changes in structure and function of cecal bacteria with early life PCB 126 exposure, we sought to explore the influence of PCB 126 on bacterial metabolism. MetaMapp networks constructed based on UHPLC-MS/MS data showed that early life PCB 126 exposure had a greater impact on bacterial hydrophilic metabolites in cecal contents from mice at the 13th week compared to 6th day after exposure (Figure 5C). A subtle increase in amino acid levels and a decrease in nucleotide levels were observed in cecal bacteria from mice at the 6th day after PCB 126 exposure (Figure 5C). Notably, PCB 126 resulted in marked increases in amino acid, nucleotide, and carbohydrate metabolism in mice at the 13th week after exposure (Figure 5C). Quantitative ^1^H NMR and assays for lipopolysaccharide (LPS) and mucin analysis were also performed to quantify the levels of target bacterial metabolites (Figure 5D,F and Appendix A). PCB 126 resulted in lower levels of cecal short chain fatty acids (SCFAs) but higher levels of cecal trimethylamine (TMA) in mice at the 13th week after exposure (Figure 5D and Appendix A). Significantly lower levels of urine hippurate were observed in mice at the 13th week after PCB 126 exposure (Figure 5E). PCB 126 also resulted in higher levels of serum LPS from mice at the 13th week after exposure (Figure 5F), a major cell wall component of Gram-negative bacteria [26]. Moreover, higher levels of fecal mucin were observed in mice at the 13th week after PCB 126 exposure (Appendix A).

## 3. Discussion

Early childhood is a susceptible age for environmental chemical exposure; however, the underlying molecular mechanisms of environmental chemical-induced disease from early life exposure is not well understood. This study aimed at investigating the short-and long-term consequences of early life environmental pollutant exposure on host and gut microbiota metabolism. We demonstrate that early life Short-Term exposure of PCB 126 has delayed effects on liver amino acid, nucleotide, and lipid metabolism as well as bile acid metabolism. We also demonstrate that early life PCB 126 exposure prominently and continuously disrupts the gut ecosystem.

The delayed changes in hepatic amino acid and nucleotide metabolism following early life Short-Term PCB 126 exposure were observed in adulthood. Five days of PCB 126 exposure resulted in dramatic liver *Cyp1a1* induction at 6 days and 13 weeks after exposure, consistent with its high AHR binding affinity [20] and considerably long half-life [14,27]. No overt acute toxicity by PCB 126 exposure was observed at either 6 days or 13 weeks after exposure, supported by no changes in blood biochemical markers, serum cytokines, or liver hematoxylin and eosin (H&E) histopathology. A significantly higher ratio of liver GSSG to GSH from mice at 13 weeks after PCB 126 exposure was observed, indicating that delayed oxidative stress was due to PCB 126 exposure. Oxidative stress impacts various biological structures including cellular membranes, lipids, proteins, and nucleic acids [28]. Consistently, we observed significant changes in amino acid and nucleotide levels in liver from mice at 13 weeks after PCB 126 exposure. Liver amino acids serve as the building blocks of proteins, can be conjugated with bile acids, and may serve as substrates for the synthesis of glucose, lipids, and anti-inflammatory molecules; therefore, abnormal metabolism of amino acids in the liver results in many diseases including fatty liver and increases risk for liver cancer [29]. Our data showed that early life PCB 126 exposure resulted in significantly lower levels of a series of amino acid including BCAAs, alanine, lysine, glutamate, tyrosine, and phenylalanine in liver from mice at 13 weeks after exposure, indicating the accelerated hepatic amino acid metabolism for energy consumption, protein synthesis, and cell proliferation that was observed in liver cancer cells [30]. This observation was consistent with another persistent organic pollutant (POP) exposure study showing significant lower levels of amino acids in liver from five days of 2,3,7,8-tetrachlorodibenzofuran (TCDF)-treated mice [31]. Moreover, increases in liver nucleotide synthesis rate for unrestrained cell proliferation and growth was also observed later in life with PCB 126 exposure, which was usually observed in cancer cells and virus-infected cells [32].

Increased hepatic lipogenesis in adulthood by early life PCB 126 exposure was observed in mice. Emerging evidence suggests that the AHR is a novel regulator of nonalcoholic fatty liver disease, and its activation can induce lipid oxidation and lipogenic pathways [33,34]. Notably, promotion of hepatic lipogenesis was observed in mice at 13 weeks after PCB 126 exposure, which is associated with persistent and dramatic AHR activation at 13 weeks after exposure. It is worthy to note that early life PCB 126 exposure did not affect metabolism at 5 weeks of age but caused dramatic changes in adulthood metabolism. This can be viewed as a special case of chronic toxicity that usually happens during critical periods of early development exposure to low levels of chemicals [12,13].

Early life PCB 126 exposure resulted in a persistent impact on the gut microbiota in adulthood. The toxicity studies are mainly observed by in vitro studies using cell lines or in vivo exposure on various experimental animal models [35]. However, there are few regulatory guidelines that address microbiome toxicity by environmental pollutant exposure. In our previous studies, we reported that POPs rapidly and significantly altered the bacterial community structural, metabolic, and transcriptional levels in both in vitro and in vivo models indicating microbial toxicity following environment pollutant exposure [25,31,36]. We observed rapid changes in microbial community structure and overall metabolism and more dramatic metabolic and gene expression changes on microbiota in adulthood by early life PCB 126 exposure. Early life PCB 126 exposure resulted in decreased abundances of genus *Akkermansia* in mice at 13 weeks after exposure, as a key beneficial mucin-degrading microbe [37,38]. The increased abundances of genus *Lachnoclostridium* were observed with PCB 126 exposure later in life, which recently has been identified as a TMA-producing bacteria [39]. This observation was consistent with the higher level of cecal TMA in mice at 13 weeks after PCB 126 exposure, which is linked to an increased risk for cardiovascular disease [40]. Early life PCB 126 exposure not only altered microbial community structure, but also significantly affected gene abundance and metabolism later in life. PCB 126 exposure resulted in significant changes in microbial amino acid, nucleotide, and energy metabolism in both childhood and adulthood. Microbial metabolism of amino acid, nucleotide, and carbohydrate in the human gastrointestinal tract plays an important role in host’s protein and lipid metabolism as well as energy homeostasis [41,42,43]. The reduction in the bacterial pathways of indigestible carbohydrate degradation and amino acid biosynthesis was observed later in life with early life PCB 126 exposure, indicating the disruption in host amino acid and energy homeostasis that may be associated with the development of obesity, insulin resistance, and diabetes [42,44]. This notion is supported by the observation of significantly higher levels of amino acids and lower levels of SCFAs derived from gut microbiota fermentation of indigestible polysaccharides [45] in the cecal content from adult mice with early life PCB 126 exposure. The profound changes in bacterial lipid metabolism were also observed in adulthood with early life PCB 126 exposure, which was consistent with the in vitro experiment showing significant impacts on bacterial lipid profiles following exposure to four different POPs [25]. This notion is supported by the observation of the disruption in the bacterial pathways of cell way biosynthesis in adulthood by PCB 126 exposure, suggesting a prominent and delayed change in bacterial membrane fluidity in response to PCB 126 [25]. Moreover, the significant changes in bile acid levels, as important host-gut microbial co-metabolism products, were observed in adulthood with early life PCB 126 exposure, suggesting the remodeling of gut microbiota and persistent AHR activation by PCB 126 [31]. Early life PCB 126 exposure also resulted in significantly lower levels of urine hippurate and higher levels of circulating LPS later in life, indicating the decreased gut microbiome diversity and Gram-negative bacterial infection that are associated with various negative health effects [46,47]. Together, these findings suggest possible new avenues for probing microbial toxicity as a key target of early life environment pollutant exposure.

## 4. Materials and Methods

### 4.1. Animals and Diets

Animal experiments were performed using protocols approved by the Pennsylvania State University Institutional Animal Care and Use Committee (PROTO202001416). Male C57BL/6J wild-type mice (three-week-old) were obtained from Jackson Laboratory (Bar Harbor, MN). After acclimatization, mice were trained to eat transgenic bacon-flavored dough pills that were prepared with tablet mold (Bio-Serve, Flemington, NJ, USA). After training for five days, mice were fed with the dough pills containing PCB 126 (a final dose of 24 µg/kg) or acetone as vehicle continuously for five days (one pill per mouse per day). Two time points were used to evaluate the short- and long-term effects of early life PCB 126 exposure on mice: (i) the mice (six mice per group) were sacrificed on the day after last PCB 126 exposure (6th day); (ii) the mice (six mice per group) were sacrificed at 13th week after PCB 126 exposure (13th week). The body weight of mice was monitored during the experiment. Tissue samples were harvested at the end of the experiment for microbial and metabolomics analyses.

### 4.2. Histological and Immunohistochemical Analyses

Paraffin-embedded liver blocks were sectioned and stained with hematoxylin and eosin (H & E). Rat monoclonal anti-mouse macrophage F4/80 (1:100, Invitrogen, Carlsbad, CA, USA) was visualized in frozen liver sections according to “Immunohistochemistry Protocol for Frozen Sections” (BioLegend, San Diego, CA, USA) or stained with oil red O by Histoserv, Inc. (Germantown, MD, USA).

### 4.3. Blood Clinical Biochemistry and Cytokine Analysis

Common liver enzymes including alanine transaminase (ALT) and alkaline phosphatase (ALP) were measured using the VetScan VS2 Chemistry Analyzer and the Mammalian Liver Profile rotor (Abaxis Inc., Union City, CA, USA). Levels of serum cytokine were measured using a BioPlex 200 mouse cytokine array/chemokine array 32-Plex by Eve Technologies (Calgary, AB, Canada).

### 4.4. Fecal Mucin and Serum Lipopolysaccharide Quantification

Fecal mucin levels were determined with Fecal Mucin Assay Kit (Cosmo Bio USA, Carlsbad, CA, USA). Serum lipopolysaccharide (LPS) levels were measured with Pierce^TM^ LAL Chromogenic Endotoxin Quantitation Kit (Thermo Fisher Scientific, Waltham, MA, USA).

### 4.5. Liver Triglyceride and Glutathione Quantification

Liver triglyceride was determined using Triglyceride Colorimetric Assay Kit (Cayman Chemical, Ann Arbor, MI, USA). The levels of reduce glutathione (GSH) and oxidized glutathione (GSSG) were quantified using GSH/GSSG Ratio Detection Assay Kit (Abcam, Cambridge, UK).

### 4.6. Tissue RNA Isolation and Quantitative PCR

RNA was extracted from frozen liver and intestine tissues using TRIzol reagent (Invitrogen, Carlsbad, CA). The cDNA was synthesized from total RNA using qScript cDNA SuperMix (Quanta Biosciences, Gaithersburg, MD, USA). Quantitative PCR (qPCR) reactions were performed on a QuantStudio 3 Real-Time PCR system (Thermo Fisher Scientific, Waltham, MA, USA). Gene-specific primers were listed in Appendix A. All results were normalized to *Gapdh* mRNA according to ΔΔC_T_ method.

### 4.7. ^1^H NMR Based Metabolomics Experiments

Biological sample preparation for NMR analyses were performed using procedure described previously [48,49]. ^1^H NMR spectra were recorded at 298 K on a Bruker Avance NEO 600 MHz spectrometer equipped with an inverse cryogenic probe (Bruker Biospin, Ettlingen, Germany). All 1D NMR spectra were acquired employing the first increment of NOESY pulse sequence (NOESYPR1D). The metabolites were assigned on the basis of published results [48,49] and further confirmed with 2D NMR spectra. ^1^H NMR spectra for liver lipids were corrected for phase and baseline distortions manually with the chemical shift referenced to TMS (*δ* = 0.0) using TopSpin 3.6 (Bruker Biospin, Ettlingen, Germany). The spectral region was integrated with AMIX 3.9 (Bruker Biospin, Ettlingen, Germany) and quantified for lipid classed in liver, as previously reported [49]. The spectra process and quantification for liver hydrophilic, cecal content, and urine metabolites were processed using Chenomx NMR Suite (Chenomx Inc., Edmonton, AB, Canada). Heatmaps were created with RStudio (pheatmap), version 1.0.12.

### 4.8. Bile Acid Quantitation by UHPLC-MS/MS

Bile acid quantitation was acquired with an ACQUITY UHPLC system using an ACQUITY C8 BEH UPLC column (2.1 mm × 100 mm, 1.7 µm) coupled with a Waters Xevo TQS mass spectrometer equipped with an electrospray ionization source operating in negative mode (all from Waters, Milford, MA, USA), as previously described [50]. Briefly, 25 mg of liver or fecal samples were homogenized in 1 mL of pre-cooled methanol containing 0.5 µM of deuterated internal standards (Sigma-Aldrich, St Louis, MO, USA and Cayman Chemical, Ann Arbor, MI, USA), followed by centrifugation. Multiple reaction monitoring for conjugated bile acids and selected ion monitoring for unconjugated bile acids were detected. The data were normalized to their respective deuterated internal standards and quantified by comparing integrated peaks against a standard curve.

### 4.9. LC-MS Based Metabolomics Analysis

The cecal content (25 mg) were added with 1 mL of pre-cool 80% methanol containing 0.1% formic acid, followed with homogenization and centrifugation. The supernatants were dried in a vacuum and resuspend in 500 μL of 3% methanol containing 1 μM chlorpropamide. Hydrophilic metabolite profiling was performed with a Dionex Ultimate 3000 quaternary HPLC system connected to Exactive^TM^ Plus Orbitrap mass spectrometer (Thermo Fisher Scientific, Waltham, MA, USA) with a Waters XSelect HSS T3 column (2.1 mm × 100 mm, 2.5 µm). The LC-MS data were analyzed with software pipeline MS-DIAL [51]. Biochemical and chemical similarities among identified hydrophilic metabolite were calculated and plotted using MetaMapp network analysis as previously described [52].

### 4.10. 16S rRNA Gene Sequencing Analysis

DNA from cecal contents were extracted using E.Z.N.A. stool DNA kit (Omega Bio-Tek Inc., Norcross, GA, USA). Aliquots were taken from all the samples to make the final concentration 10 ng/µL. Then, V4 region of the 16S rRNA gene of the bacteria was amplified using the primer set 515F and 806R. Amplicons were checked for band size of 292 bp using 1.5% agarose gel electrophoresis with DNA 7500LabChip on the Agilent 2100 Bioanalyzer (Agilent Technologies, Santa Clare, CA, USA). Amplified DNA samples were submitted to Pennsylvania State University Genomics Core Facility (University Park, PA) for 250 × 250 paired end Illumina Miseq sequencing. Obtained raw data were analyzed using mothur platform [53] and SILVA v138 database [34].

### 4.11. Metagenomic Analysis

For shotgun metagenomics, the cecal DNA samples were submitted to Pennsylvania State University Genomics Core Facility (University Park, PA, USA) for NextSeq Mid-Output 150 × 150 paired end sequencing. Obtained demultiplexed reads underwent quality trimming and adaptor using FastQC (https://www.bioinformatics.babraham.ac.uk/projects/fastqc/, accessed on 5 July 2022) and host read removal using Kneaddata [54]. Clean metagenomic sequence reads were analyzed using Kraken2 taxonomic sequence classification approach on standard Kraken database comprising of all complete bacterial, viral and archeal genomes in RefSeq [55]. Abundance of the various species was estimated using Bracken [56]. For functional classification, reads were concatenated and then processed with default settings using HUMAnN3 [57].

### 4.12. Statistics

Graphical illustrations and statistical analyses were performed using GraphPad Prism 6.0 (GraphPad, San Diego, CA). All data values are presented as mean ± standard deviation (SD) or median and interquartile range. The data were analyzed using unpaired *t* test analyses and *p* < 0.05 were considered as significant.

## 5. Conclusions

Collectively, these data demonstrate that early life PCB 126 exposure has long-term consequences on liver amino acid and nucleotide metabolism as well as increased hepatic lipogenesis later in life. We also determined a persistent disruption in gut microbiota later in life by early life PCB exposure. These results find significant association between early life environmental pollutants exposure and abnormal metabolism later in life and suggest the microbiome is a key target of environmental chemical exposure.

## Figures and Tables

**Figure 1 ijms-23-08220-f001:**
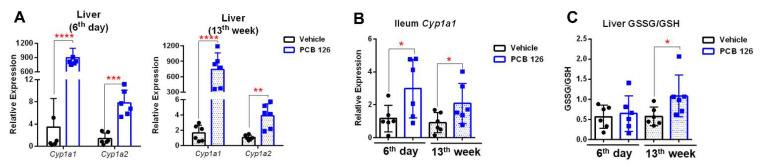
Effects of early-life PCB 126 exposure on AHR signaling and redox status. (**A**,**B**) qPCR analysis of mRNA levels of AHR target genes in the liver (**A**) and ileum (**B**) from mice with vehicle or PCB 126 exposure. (**C**) Ratio of oxidized glutathione (GSSG) to reduced glutathione (GSH) in the liver from mice with vehicle or PCB 126 exposure. Values are means ± S.D. (*n* = 6 per group). * *p* < 0.05, ** *p* < 0.01, *** *p* < 0.001, **** *p* < 0.0001 compared to vehicle.

**Figure 2 ijms-23-08220-f002:**
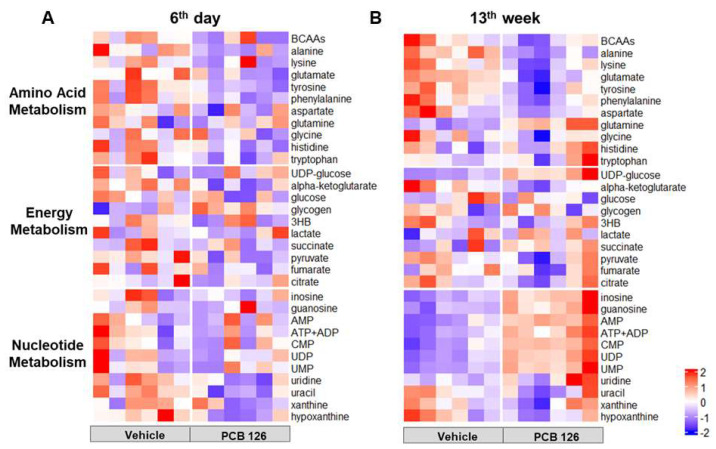
Effects of early life PCB 126 exposure on liver global metabolism. Heat map representation of relative content of liver hydrophilic metabolites obtained from ^1^H NMR analysis from mice with vehicle or PCB 126 exposure at the 6th day (**A**) or 13th week (**B**). *n* = 6 per group. BCAAs, branched-chain amino acids; UDP-glucose, uracil-diphosphate glucose; 3HB, 3-hydroxybutyric acid; AMP, adenosine monophosphate; ADP, adenosine diphosphate; ATP, adenosine triphosphate; CMP, cytidine 5′-monophosphate; UDP, uridine 5′-diphosphate; UMP, uridine monophosphate.

**Figure 3 ijms-23-08220-f003:**
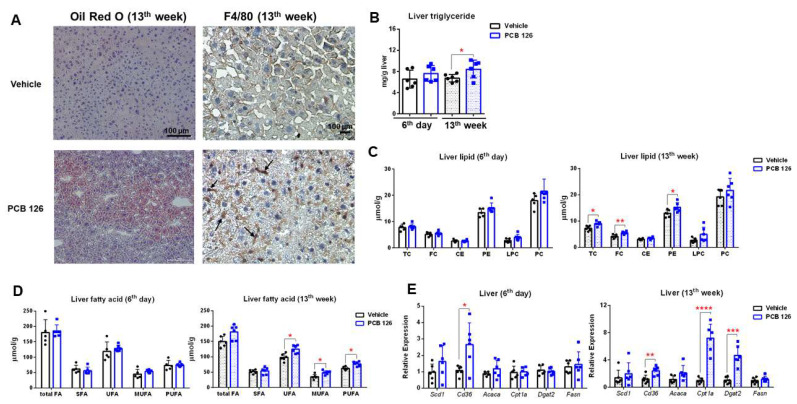
Effects of early life PCB 126 exposure on hepatic lipid metabolism. (**A**) Oil Red O and F4/80-positive cells staining in the liver from mice with vehicle or PCB 126 exposure. (**B**) Liver triglyceride levels from mice with vehicle or PCB 126 exposure. (**C**,**D**) Quantitative NMR analysis of liver lipid (**C**) and fatty acid (**D**) profiling from mice with vehicle or PCB 126 exposure. (**E**) qPCR analysis of mRNA encoding de novo fatty acid biosynthesis in liver from mice with vehicle or PCB 126 exposure. Values are means ± S.D. (*n* = 6 per group). * *p* < 0.05, ** *p* < 0.01, *** *p* < 0.001, **** *p* < 0.0001 compared to vehicle. TC, total cholesterol; FC, free cholesterol; CE, cholesterol ester; PE, phosphatidylethanolamine; LPC, lysophosphatidylcholine; PC, phosphatidylcholine; FA, fatty acid; SFA, saturated fatty acid; UFA, unsaturated fatty acid; MUFA, monosaturated fatty acid; PUFA, polyunsaturated fatty acid.

**Figure 4 ijms-23-08220-f004:**
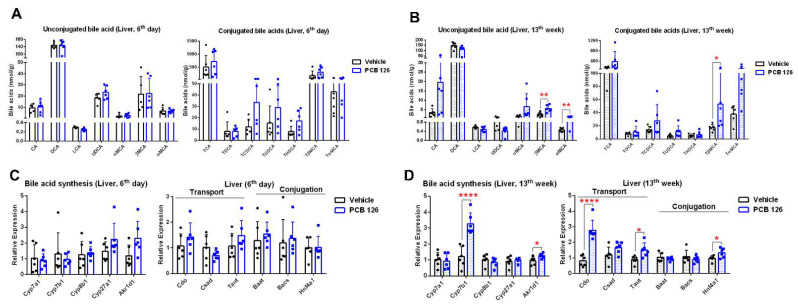
Effects of early life PCB 126 exposure on liver bile acid metabolism. (**A**,**B**) Quantitative UHPLC-MS/MS analysis of bile acids in the liver from mice with vehicle or PCB 126 exposure. (**C**,**D**) qPCR analysis of mRNA encoding bile acid biosynthesis, transport, and conjugation in the liver from mice with vehicle or PCB 126 exposure. Values are means ± S.D. (*n* = 6 per group). * *p* < 0.05, ** *p* < 0.01, **** *p* < 0.0001 compared to vehicle. CA, cholic acid; DCA, deoxycholic acid; LCA, lithocholic acid; UDCA, ursodeoxycholic acid; MCA, muricholic acid; T, taurine-conjugated.

**Figure 5 ijms-23-08220-f005:**
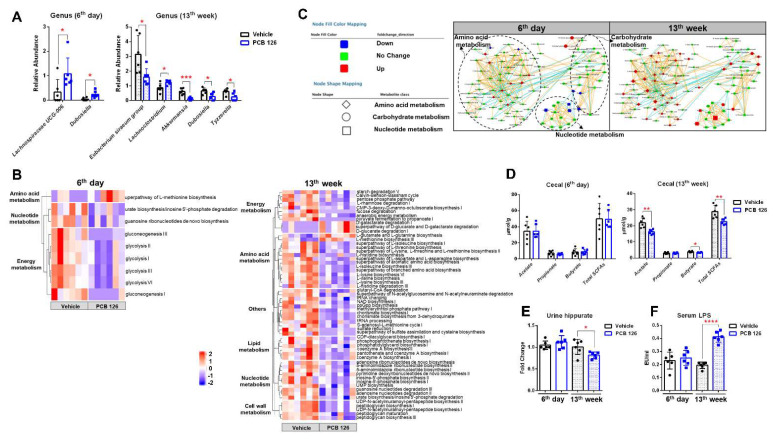
Effects of early life PCB 126 exposure on bacteria community, gene expression, and metabolism. (**A**) Relative abundance of cecal bacteria from genus that significantly changed from mice with vehicle or PCB 126 exposure. (**B**) Analysis of KEGG pathway abundance that significantly changed in the cecal microbiota from mice with vehicle or PCB 126 exposure. (**C**) Metabolic network changes in cecal bacteria from mice with vehicle or PCB 126 exposure. (**D**,**E**) NMR analysis of cecal short-chain fatty acids (SCFAs) (**D**) and urine hippurate (**E**) from mice with vehicle or PCB 126 exposure. (**F**) Serum lipopolysaccharide (LPS) levels from mice with vehicle or PCB 126 exposure. Values are means ± S.D. (*n* = 6 per group). * *p* < 0.05, ** *p* < 0.01, *** *p* < 0.001, **** *p* < 0.0001 compared to vehicle.

## Data Availability

The data presented in this study are available on request from the corresponding author.

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
