# Peer review of "Early Life Short-Term Exposure to Polychlorinated Biphenyl 126 in Mice Leads to Metabolic Dysfunction and Microbiota Changes in Adulthood"

_ijms, 2022, doi:10.3390/ijms23158220_

Round 1

Reviewer 1 Report

In the present study the authors investigate the early-life exposure to environmental pollutants using the example of short-term exposure of mouse pups to 3,3',4,4',5-pentacholorobiphenyl. They found that the exposure had the short-  and long-term consequences of early-life environmental pollutant exposure on host metabolism and its gut microbiota metabolism. This interesting, important and well-presented article can be accepted for publication.

The main question addressed by the research is provides evidence for an association between early-life environmental pollutant exposure and risk of metabolic disorders later in life.  The topic is relevant in the field, provides new information about the long-term consequences of early-life environmental pollutants exposure and finds association between early-life environmental exposure and abnormal metabolism in adulthood on host and his microbiome. The conclusions consistent with the evidence and arguments, the references are appropriate. The gaps in the figures (e.g. Figure 4. a and b) do not look very good, but hard to suggest how to do it better. It would be possible to suggest the authors to write a more detailed conclusion but I don't insist on it.

Author Response

We appreciate the reviewer’s positive comment. We changed Figure 4A-B from to “Box-and-Whisker plot” version to “Bar plot” version.

Reviewer 2 Report

This paper by Patterson and coworkers reports the effects of exposing mice to PCB 126. Their data provide evidence of the risks associated with this chemical in short and long term.  I enjoy reading the paper/data. The only thing I would add to this paper is the sources of this pollutant. Meaning, where and how humans are exposed to this specific compound.

Author Response

We appreciate the reviewer’s positive comment.

Good point. We added this information in the Introduction Section of the manuscript as “Animal fat tissues (such as meat and dairy products) are a major source of PCBs, since PCBs are lipophilic and highly resistant to physical, chemical, and enzymatic breakdown [15].”

Ref [15]. Faroon, O. M.; Keith, L. S.; Smith-Simon, C.; Rosa, C. T. D. Polychlorinated biphenyls : human health aspects; 2003; World Health Organization; pp 1-64.

Reviewer 3 Report

The manuscript (ijms-1829056) titled "Early-life short term exposure to polychlorinated biphenyl 126 in mice leads to metabolic dysfunction and microbiota changes in adulthood" is well written, and all the experimental data well presented and discussed. I think it will interest the International Journal of Molecular Sciences readers. It is also within the journal's scope and suitable for publication.

As far as I could notice, the figures are all precise, mainly the graphics. I did not detect any inappropriate references. All seemed to be essential and correct for the situation where the authors used them. I did not detect problems in the experimental procedures. When they were not fully explained, proper references were used. I could find all the information necessary to understand the procedure, and I think to reproduce it. Obvious not everyone has the conditions to make in vivo tests, but that does not mean the experiments are not well explained. The topic addressed is in continuation of the author's work, so a few points are similar to previous results. However, these days where pollution is an important issue and we face several problems, I think it is essential to have specific studies addressing the possible implications for health due to exposure to some chemicals, especially if we consider that we cannot control some because they just are released to the atmosphere. So I think the submitted manuscript is relevant, so I suggest its publication. Overall, I found the manuscript interesting for other researchers in the same field and other fields. Naturally, this is just my opinion. I did like the article and learned a few things about it.

Author Response

We appreciate the reviewer’s positive comment.